# Identification of outcomes to inform the development of a core outcome set for surgical innovation: a targeted review of case studies of novel surgical devices

Nicholas Wilson,[1] Rhiannon C Macefield [ID],[1] Christin Hoffmann [ID],[1] Matthew J Edmondson,[2] Rachael L Miller [ID],[1,3] Emily N Kirkham,[4] Natalie S Blencowe [ID],[1,5] Angus G K McNair [ID],[1,6] Barry G Main [ID],[1,5,7] Jane M Blazeby [ID],[1,5] Kerry N L Avery [ID],[1] Shelley Potter [ID] [1,8]

NW and RCM are joint first authors.

KNLA and SP are joint senior authors.

For numbered affiliations see end of article.

**Correspondence to**
Rhiannon C Macefield;
r.macefield@bristol.ac.uk

## ABSTRACT

**Objective** Outcome selection and reporting in studies of novel surgical procedures and devices lacks standardisation, hindering safe and effective evaluation. A core outcome set (COS) to measure and report in all studies of surgical innovation is needed. We explored outcomes in a specific sample of innovative surgical device case studies to identify outcome domains specifically relevant to innovation to inform the development of a COS.

**Design** A targeted review of 11 purposive selected case studies of innovative surgical devices.

**Methods** Electronic database searches in PubMed (July 2018) identified publications reporting the introduction and evaluation of each device. Outcomes were extracted and categorised into domains until no new domains were conceptualised. Outcomes specifically relevant to evaluating innovation were further scrutinised.

**Results** 112 relevant publications were identified, and 5926 outcomes extracted. Heterogeneity in study type, outcome selection and reporting was observed across surgical devices. Categorisation of outcomes was performed for 2689 (45.4%) outcomes into five broad outcome domains. Outcomes considered key to the evaluation of innovation (n=66; 2.5%) were further categorised as surgeon/operator experience (n=40; 1.5%), unanticipated events (n=15, 0.6%) and modifications (n=11; 0.4%).

**Conclusion** Outcome domains unique to evaluating innovative surgical devices have been identified. Findings have been combined with multiple other data sources relevant to the evaluation of surgical innovation to inform the development of a COS to measure and report in all studies evaluating novel surgical procedures/devices.

## INTRODUCTION

The introduction and evaluation of innovative surgical procedures and devices is less regulated and standardised than that required for new pharmaceuticals. This can cause patient harm, as demonstrated by examples such as the introduction of

### Strengths and limitations of this study

► A purposive sample of surgical device case studies was selected to ensure broad representation across surgical specialties and disease areas.
► Studies were chosen to reflect the breadth of surgical device innovation.
► A broad definition of an outcome was adopted to ensure that all information potentially relevant to evaluating device innovation was captured.
► This review was limited by searching for relevant publications using device tradename only and restricting to English language publications.

metal-on-metal hip replacements and vaginal mesh implants.[1 2] One particular challenge for the evaluation of surgical innovation is the lack of standardisation in the selection, measurement and reporting of outcomes in studies of new procedures and devices, limiting data synthesis and the ability to accurately compare and contrast device efficacy.[3] The Idea, Development, Exploration, Assessment, Long-term Follow-up (IDEAL) framework describes a structure and recommended study designs for the introduction and evaluation of new surgical procedures and devices.[4 5] A consensus-agreed core set of outcomes that are essential to measure at each stage of the innovation pathway, however, is lacking from the guidance. Decisions regarding the choice of outcomes are currently made by study investigators. Outcomes may be selected a priori (based on hypothesised efficacy) or post hoc (based on statistical significance), rather than being selected because they are meaningful to patients or valuable to surgeon-innovators and other stakeholders looking to adopt the procedure. These issues risk the potential for introducing outcome reporting

BMJ

bias and inconsistent and heterogenous outcome selection which, when combined with thwarted data synthesis, may delay the identification of the benefits and harms of new interventions.[6]

A potential solution to this problem is to develop and use a core outcome set (COS) to evaluate innovative surgical procedures/devices. A COS is an agreed set of outcomes to be measured and reported, as a minimum, in all studies in a particular area.[7] Use of COS has been shown to be effective at improving the quality and consistency of outcome selection and reporting in comparative effectiveness research, for example, randomised controlled trials (RCTs).[8] Methods for developing COS for effectiveness studies for particular diseases or conditions are well established. This includes, first, the identification of a 'long list' of all potential outcomes to inform the consensus process and reduce to a core set, typically by undertaking systematic reviews of RCTs in the specific disease or condition to identify outcomes of interest.[7] This is frequently followed by a Delphi consensus process involving key stakeholders to prioritise a minimum set of core outcomes that are essential to include in the COS. However, systematic reviews of RCTs are unlikely to identify outcomes of specific relevance to surgical innovation. Furthermore, important 'drivers' (ie, factors that may influence the development) of innovation may not be recognised and reported as outcomes in the traditional sense and instead may be reported in more descriptive terms. Novel approaches are needed. We hypothesised that publications reporting the introduction and evaluation of new surgical procedures and devices may generate specific insights into outcome selection and reporting in surgical innovation, and thereby serve as valuable data sources to generate a long list of potential outcomes to inform the development of a future COS. The intention of this initial targeted review was to focus on surgical devices.

The aim of this study was to (1) examine outcome selection and reporting in a purposive sample of case studies of known innovative surgical devices; (2) categorise outcomes into domains, identifying outcomes specifically relevant to innovation; and (3) conceptualise innovation domains for inclusion in a future Delphi consensus process to develop a COS.[9 10] This study, focusing on studies of surgical devices, served as the first of multiple other data sources of relevance to surgical innovation to conceptualise outcome domains for inclusion in the COS.[9]

## METHODS

A targeted review of purposively selected case studies of innovative surgical devices was undertaken. A purposive sampling approach (detailed below) was undertaken with the aim of representing, and being able to examine in depth, a broad and varied range of innovative surgical devices.

### Selection of case studies

The purposive sampling strategy was discussed and agreed among members of the multidisciplinary study team. Specialists from a range of medical disciplines (orthopaedic surgery, neurosurgery, plastic surgery, gynaecology, gastrointestinal surgery, breast surgery, cardiology, urology, vascular surgery, maxillofacial surgery and ophthalmology) known to the study team were contacted and asked to identify innovative surgical devices developed within the last decade. A purposive sample of the identified devices were then selected by the study team to ensure broad representation across the surgical specialties, disease areas and 'degree of novelty' of the innovation, to ensure that case studies reflected the breadth of the surgical device innovation life cycle. Degree of novelty was categorised through discussion within the study team as 'wholly innovative' (a device representing a completely new approach to solving a clinical problem); 'partially innovative' (a device broadly similar in function to one already in use but differing in at least one significant way); 'reinvented' (a modification of a device or technique that had been previously used but that was abandoned due to complications); or 'predicate' (a device introduced via the US Food and Drug Administration (FDA) predicate 510(k) pathway on the basis of equivalence[11]).

### Identification and selection of publications

Electronic searches were performed in PubMed to identify publications reporting the introduction and evaluation of each device in July 2018. Searches were performed using the tradename of the device as a text word (online supplemental file 1). No publication date limits were applied. Search results were imported into Excel. Records were screened for eligibility by one reviewer with 10% independently checked for eligibility by a second reviewer. Included were primary studies reporting the use of the device at any stage of development from first in human up to and including RCTs. Excluded were studies involving animals and publications in non-English languages. Letters and conference proceedings were also excluded due to lack of detail to extract comprehensive information on outcome selection.

### Data extraction

Descriptive characteristics of included publications, including publication year, study details (type of study, design, number of centres involved and whether they had ethical/institutional review board approval) and geographical origin, were extracted. Outcome data extraction for each publication was approached systematically starting with the title. Outcomes were extracted verbatim on first mention in the publication. Data extraction included generic descriptions of outcomes (eg, safety outcomes) as well as detailed, specific descriptions (eg, specific types of adverse events). For the purpose of this review, a broad definition of an outcome was adopted to ensure that all information potentially relevant to the evaluation of innovation was captured.

This included any measured or reported construct or concept relating to or occurring as a result of using the device.[6] It also included descriptions or details of events and observations that occurred during the course of the study. Outcomes that were discussed (but not necessarily measured and reported as a study endpoint) by the publication's authors as being of interest or of relevance to evaluating the device were also extracted.

For each device case study, data extraction was performed on subsequent publications until no new outcomes were seemingly being observed. The order in which publications were selected for data extraction within each case study was not specified, irrelevant of publication date or study type. Data extraction was performed by one reviewer (either NW, RCM, KA, RLM, ENK or MJE) with double data extraction completed on 10% of papers. Any discrepancies were discussed and resolved within the study team. Data were extracted directly into an electronic purpose-designed database using Research Electronic Data Capture software.[12]

### Data analysis: categorisation of outcomes into domains

The complete list of verbatim outcomes extracted from all publications was reviewed by members of the study team with clinical knowledge and methodological expertise in COS development (SP, KA, RCM, NW). Outcomes were categorised into a conceptual framework of outcome domains (broad classifications of aspects relating to the effects or use of the device), referring back to source publications for context if needed. The conceptual framework of outcome domains was informed by the experience of the study team and knowledge of the existing taxonomies of outcomes traditionally measured in effectiveness studies.[13] During the progression of outcome categorisation, the framework of conceptualised domains was iteratively refined, informed by the data and discussion within the study team. Domains included, for example, adverse events (ie, unfavourable signs, symptoms or incidents), clinical efficacy (eg, indications of success in treating the condition) and patient-reported outcomes. During analysis, outcomes that were considered as specific to the evaluation of innovation in the context of the included case studies (ie, were particularly relevant for considering the future use, modification or uptake/abandonment of the device) were categorised as 'key to evaluation of innovation'. Categorisation of outcomes into domains continued until the list of domains was comprehensive and it was possible to categorise all outcomes into the conceptualised domains. For the purpose of this review, outcomes that were categorised as 'key to evaluation of innovation' were scrutinised in more detail by the study team and further categorised into subdomains, referring to the original publication for verification of context. These subdomains were not predefined, and categorisation was iterative and data led.

Findings were summarised using descriptive statistics, where appropriate, and narrative summaries of the outcome domains relevant to evaluating innovation.

### Patient and public involvement

This review was one component of a wider study to develop a COS for surgical innovation which sought patient and public input in the concept, design, analysis and dissemination at all stages of the study.[9] This was achieved through study steering group meetings with patient and public representatives and through two National Institute for Health and Care Research (NIHR) Bristol Biomedical Research Centre (BRC) Surgical Innovation theme patient advisory group meetings. Patients and the public were not directly involved in conducting the review or performing the analysis.

## RESULTS
### Device case studies and search results

Eleven device case studies were identified to reflect the breadth of the device innovation life cycle. Details of each device, their degree of novelty and brief descriptions of their functions are provided in table 1.

Electronic database searches yielded 242 records, of which 114 (47%) were excluded (figure 1). Full-text articles of the remaining 128 records were examined for eligibility. All reported on the use and/or evaluation of the device under study and were included for review (online supplemental file 2).

### Data extraction

In total, data extraction was completed on 112/128 (87.5%) of all included publications (figure 1). This included all publications for 10 of the 11 device case studies. The exception was the Activa Deep Brain Stimulation device. This was the last case study for which data extraction was conducted. After data extraction from 11 of the 27 included publications, no new outcomes were observed to be emerging and the study team agreed that further extraction would not likely add to the conceptualisation of outcome domains. The remaining 16 publications identified for this case study were, therefore, excluded from further analysis.

### Publication characteristics

Descriptive details of the analysed publications, presented by device case study, are displayed in table 2. Publication dates ranged from 2009 to 2018. Most publications were from North America (n=49; 43.8%), followed by the European Economic Area (n=25; 22.3%). Publications were predominantly case series including non-comparative cohort studies (n=60; 53.6%). Other publications were single-patient case reports (n=24; 21.4%), non-randomised comparative studies (n=19; 17.0%) and a small number of RCTs (n=9; 8%). Just over half of the publications reported data from a single centre (n=61; 54.5%) and approximately a third reported data from multicentre studies (n=40; 35.7%). Some 11 publications (9.8%) did not report the number of centres involved. Publications reported retrospective (n=62; 55.0%) and prospective (n=50; 45.0%) data collection. A statement

**Table 1** Description of the 11 medical devices selected to explore outcome selection and reporting

| Degree of novelty* | Device tradename | Licensed by | Surgical specialty | Description of device |
|---|---|---|---|---|
| Wholly innovative | Activa Deep Brain Stimulation | Medtronic | Neurosurgery | A dual-channel, rechargeable neurostimulator for treatment of neurological conditions including Parkinson's disease, dystonia and essential tremor. The device is typically implanted in the chest or abdomen, connected to an extension and leads, which are implanted in the brain. |
| | Magseed | Endomag | Breast | A ferromagnetic localisation seed used to localise impalpable breast lesions prior to excision. |
| | UroLift | Neotract | Urology | A minimally invasive treatment of benign prostatic hyperplasia or an enlarged prostate. |
| Partially innovative | AeroForm | AirXpanders | Plastic/breast | A remote-controlled tissue expander filled from an internal source with carbon dioxide eliminating the need for needle injections to fill the expander. This expander makes breast reconstruction faster and more comfortable while providing women with some degree of control over the expansion. |
| | Micra leadless pacemaker | Medtronic | Cardiology | A pacemaker implanted directly into the patient's heart. |
| | Zenith fenestrated AAA graft | Cook Medical | Vascular | An endovascular graft indicated for the endovascular treatment of patients with abdominal aortic or aortoiliac aneurysms of a morphology not suitable for standard endovascular repair. |
| Reinvented | Braxon | Raise Healthcare | Plastic/breast | A preshaped porcine acellular dermal matrix designed to completely enclose the breast implant prior to fixation on top of the intact pectoralis major muscle for breast reconstruction. |
| | LINX Reflux Management System | Ethicon (formally Torax Medical) | Gastrointestinal | An antireflux device that can be inserted laparoscopically around the lower end of the oesophagus for the treatment of gastro-oesophageal reflux disease. |
| Predicate | Accolade II hip stem | Stryker | Orthopaedics | A morphometric wedge femoral hip stem, merging conventional tapered wedge femoral stem design with size-specific medial curvature to more closely fit a broad range of bone sizes and shapes of today's patient population. |
| | Align urethral support system | Bard Medical | Urology/gynaecology | A macroporous lightweight polypropylene mesh designed to treat cases of stress urinary incontinence as well as pelvic organ prolapse. |
| | BioDesign fistula plug | Cook Medical | Gastrointestinal | A fistula plug for implantation to reinforce soft tissue for the repair of rectovaginal or anorectal fistulas. |

*Wholly innovative: a device that represents a completely new approach to solving a clinical problem. Partially innovative: a device that is broadly similar in function to one already in use but differs in at least one significant way. Reinvented: a modification of a device or technique that was previously abandoned for complications. Predicate: a device introduced via the Food and Drug Administration (FDA) predicate 510(k) pathway on the basis of equivalence.

confirming ethical or institutional review board approval was reported in 72 (64.3%) publications.

### Categorisation of outcomes into domains

A total of 5926 outcomes were extracted from the 112 included publications (table 2). The number of extracted outcomes for each device case study ranged from 117 individual outcomes (Magseed device; extracted from a total of two publications) to 1320 outcomes (LINX Reflux Management System; extracted from a total of 22 publications). The median number of outcomes extracted from each unique publication was 48, ranging from 5 to 135.

Outcome categorisation continued until 2689 (45.4%) outcomes had been categorised to derive a conceptual framework of five broad outcome domains. The conceptual framework included: (1) anticipated or procedural events (defined as expected or routine incidents or consequences, n=1192; 44.3%), (2) clinical, technical or common data element parameters (defined as routinely measured data variables of interest to the clinical specialty,[14] n=745; 27.7%), (3) device function (defined as technical or operational performance parameters specific to the device, n=446; 16.6%), (4) patient-reported outcomes (n=240; 8.9%), and (5) outcomes key to evaluating innovation (ie, considered to be particularly relevant to the future use, modification or uptake/abandonment of the device, n=66; 2.5%) (figure 2). The remaining 3237 (54.6%) outcomes were not categorised as the list of conceptualised domains was agreed by the study team to be comprehensive and exhaustive (ie, it was possible to categorise subsequent outcomes into the

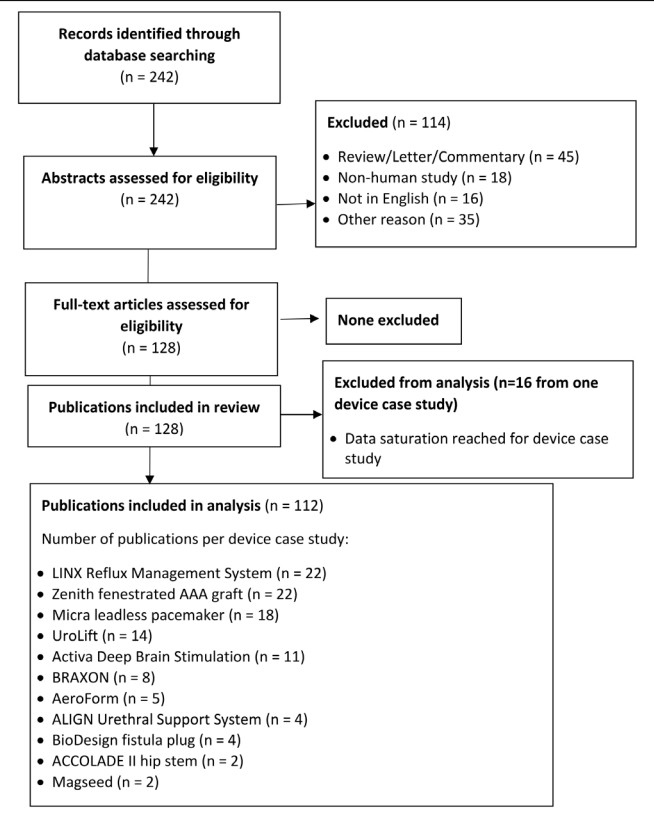

**Figure 1** Identification and selection of included publications.

existing domains) or because outcomes were too broad in their description to make categorisation possible (eg, 'safety').

## Outcomes key to evaluating innovation

Outcomes categorised as key to the evaluation of innovation (n=66; 2.5%) were further categorised into three subdomains: surgeon/operator experience, unanticipated events and modifications.

Some 40 (1.5%) outcomes were categorised as relating to surgeon/operator experience. These included, for example, ease of using the new device, convenience and satisfaction. For the majority of the included case studies, the operator of the device was the surgeon. In the case study of the AeroForm tissue expander, the device was operated by the patient following discharge from hospital. An example of an outcome relevant to patients' experience of operating the device included 'very easy' use of the dosage controller.[15]

Some 15 (0.6%) outcomes were categorised as unanticipated events. These included adverse events related to the use of the device that were reported by the authors as unexpected or not previously encountered or anticipated. An example of an outcome coded as an unanticipated event was inability to dislodge the device in a retrieval attempt in the case of the leadless cardiac pacemaker (Micra LCP).[16] A further example included fatal intestinal injury during surgery in the case study of the Align urethral support system.[17]

Some 11 (0.4%) outcomes were categorised as modifications. These included adjustments to the device, for example, redesign of the internal membrane of the Aero-Form tissue expander following testing during its development.[18] The authors noted how the expander bulk was decreased to minimise the risk of underexpansion and erosion, and therefore potential harm, in subsequent patients.[18] Details on modification outcomes, however, were rarely reported in any of the other device case studies. Of the 11 modification outcomes, seven were from the AeroForm case study with the remaining four extracted from only three of the other case studies. Modifications were often described by the authors in general terms and without specific descriptions, such as 'design modifications' in the example of the Accolade hip stem[19] and 'modifications to the liner' or 'midtrial modifications to the expander' in the example of the AeroForm device.[20]

## DISCUSSION

This review identified a broad range of outcomes identified, selected and reported in studies describing the introduction and evaluation of a purposive sample of novel devices, chosen to reflect the breadth of surgical device innovation. Although relatively few in number, outcomes considered key to the evaluation of innovation were identified from these empirical case studies, distinct from most outcomes shared with effective studies.[13] These were broadly categorised into outcome domains 'surgeon/operator experience', 'unanticipated events' and 'modifications'. Although infrequently reported in the studies included in this review, we consider these types of outcomes to be important drivers of innovation at the early stage of the device life cycle and key to assessing procedures with new medical devices.

Recent systematic reviews have demonstrated that outcome reporting in studies of innovative surgical procedures and devices focuses on short-term, clinical and technical outcomes.[21 22] These outcomes are critical to assess in any early-phase study and are required for US FDA and the UK Medicines and Healthcare products Regulatory Agency approval.[23 24] These types of outcomes are also the recommended focus of early-phase studies (stages 1 and 2) in the IDEAL framework for the introduction and evaluation of new surgical procedures, and devices (IDEAL-D).[4 5] The current review, however, has studied outcome selection and reporting in depth in empirical case studies, in order to identify innovation-specific outcomes potentially important to evaluating the process of innovation that have not explicitly been specified as key outcomes in existing guidance. Assessing surgeon/operator experience of using a new device, for example, can highlight issues and provide valuable feedback for the device developers and clinical colleagues, playing an important role in making improvements in future versions of the device or the way in which the device is inserted or used. User experience may also determine

**Table 2** Publication characteristics and number of outcomes extracted for the 11 case studies of innovative medical devices

| Degree of novelty* | Overall | Wholly innovative | | | Partially innovative | | | Reinvented | | Predicate | | |
|---|---|---|---|---|---|---|---|---|---|---|---|---|
| | | Activa Deep Brain Stimulation | Magseed | UroLift | AeroForm | Micra leadless pacemaker | Zenith fenestrated AAA graft | Braxon | LINX Reflux Management System | Accolate II hip stem | Align urethral support system | BioDesign fistula plug |
| Publications included (n) | 112 | 11 | 2 | 14 | 5 | 18 | 22 | 8 | 22 | 2 | 4 | 4 |
| Year of publication (range) | 2009–2018 | 2017–2018 | 2018 | 2013–2018 | 2014–2017 | 2015–2018 | 2016–2018 | 2016–2018 | 2010–2018 | 2014–2018 | 2013–2015 | 2009–2015 |
| Country of origin, n (%) | | | | | | | | | | | | |
| EEA | 25 (22.3) | 3 | 0 | 2 | 0 | 3 | 4 | 4 | 5 | 0 | 2 | 2 |
| North America | 49 (43.8) | 6 | 1 | 3 | 3 | 2 | 15 | 0 | 15 | 2 | 2 | 0 |
| UK | 12 (10.7) | 1 | 1 | 3 | 0 | 1 | 2 | 3 | 0 | 0 | 0 | 1 |
| Multiple | 17 (15.2) | 0 | 0 | 5 | 0 | 10 | 0 | 0 | 2 | 0 | 0 | 0 |
| Other | 9 (8.0) | 1 | 0 | 1 | 2 | 2 | 1 | 1 | 0 | 0 | 0 | 1 |
| Type of publication, n (%) | | | | | | | | | | | | |
| Case report | 24 (21.4) | 3 | 0 | 2 | 0 | 7 | 3 | 1 | 7 | 0 | 1 | 0 |
| Case series | 60 (53.6) | 6 | 2 | 6 | 1 | 2 | 17 | 7 | 11 | 2 | 2 | 4 |
| Non-randomised comparative study | 19 (17.0) | 2 | 0 | 1 | 2 | 9 | 2 | 0 | 3 | 0 | 0 | 0 |
| Randomised controlled trial | 9 (8.0) | 0 | 0 | 5 | 2 | 0 | 0 | 0 | 1 | 0 | 1 | 0 |
| Multi/single centre, n (%) | | | | | | | | | | | | |
| Single centre | 61 (54.5) | 8 | 1 | 5 | 3 | 7 | 15 | 5 | 10 | 1 | 4 | 2 |
| Multicentre | 40 (35.7) | 2 | 1 | 6 | 2 | 11 | 5 | 2 | 9 | 0 | 0 | 2 |
| Not reported | 11 (9.8) | 1 | 0 | 3 | 0 | 0 | 2 | 1 | 3 | 1 | 0 | 0 |
| Data collection, n (%) | | | | | | | | | | | | |
| Retrospective | 62 (55) | 6 | 0 | 8 | 0 | 8 | 15 | 4 | 15 | 0 | 3 | 3 |
| Prospective | 50 (45) | 5 | 2 | 6 | 5 | 10 | 7 | 4 | 7 | 2 | 1 | 1 |
| Ethical/institutional review board approved, n (%) | | | | | | | | | | | | |
| Yes | 72 (64.3) | 7 | 2 | 7 | 3 | 10 | 14 | 4 | 17 | 2 | 3 | 3 |
| No | 40 (35.7) | 4 | 0 | 7 | 2 | 8 | 8 | 4 | 5 | 0 | 1 | 1 |
| Outcomes extracted (n) | 5926 | 635 | 117 | 423 | 264 | 1012 | 1070 | 611 | 1320 | 122 | 233 | 119 |

*Wholly innovative: a device that represents a completely new approach to solving a clinical problem. Partially innovative: a device that is broadly similar in function to one already in use but differs in at least one significant way. Reinvented: a modification of a device or technique that was previously abandoned for complications. Predicate: a device introduced via the Food and Drug Administration (FDA) predicate 510(k) pathway on the basis of equivalence.
EEA, European Economic Area.

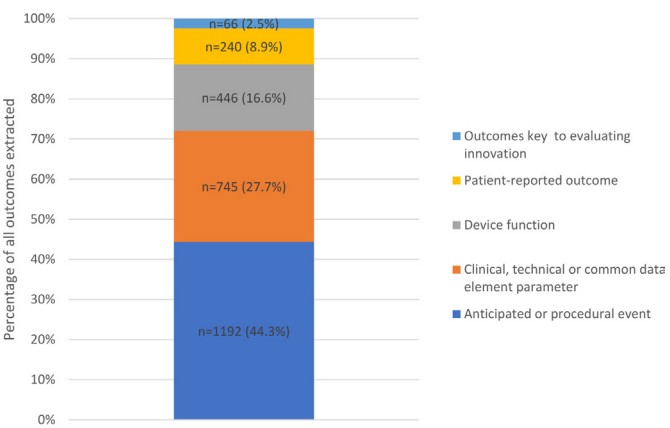

**Figure 2** Categorisation of outcomes extracted from all publications (n=2689).

whether a device is adopted into practice or abandoned, further underlining the relevance of this outcome as a key driver to innovation. The examples of unanticipated events identified in this review are also vitally important to the introduction and evaluation of innovation as they may represent patient safety issues that should be shared with clinicians undertaking the procedure in the future to minimise patient harm. It is recognised that unanticipated events are commonly reported in later phase effectiveness studies such as RCTs, in particular in studies evaluating pharmaceuticals. In the context of early-phase studies of surgical devices, however, they are particularly important to highlight unexpected problems requiring modifications to the device or the procedure for using or inserting it. Such modifications may improve the efficacy of the device, its ease of use or, perhaps most importantly, reduce the risk of patient harm. Reporting unanticipated outcomes and modifications in early stages of the introduction of a device is key to shared learning relating to innovation with the wider surgical community, providing vital information to surgeons using the device in the future and to avoid repeated mistakes or harms.

This review examined a broad range of case studies of novel devices across the innovation life cycle and across different surgical specialties. There are, however, some limitations. Case studies were identified by asking specialists known to the study team rather than a wider survey. A purposively selected sample covering a range of specialties and stage of innovation were subsequently included. It is acknowledged that the 11 included case studies are not an exhaustive sample and may not be representative of all innovative devices. This study, however, was one of many data sources to inform the list of outcomes to consider for development of the COS for surgical innovation, which used triangulated data from published literature, qualitative interviews and regulatory body documents that focused on surgical procedures and/or devices.[9] Identification of publications involved database searches performed using the device tradename as the key search term. It is recognised that some relevant studies conducted before the tradename was established

may have been missed. It is also acknowledged that a small number of outcomes may have been missed during data extraction, with only 10% of papers undergoing a second reviewer independent check. It is unlikely, however, that this would have altered the conceptualised framework of outcome domains. During analysis, a broad approach to categorise outcomes was taken, using a conceptual framework that evolved during the iterative process of data analysis. Categorisation of outcomes during the earlier stages of analysis, therefore, may not represent how they may have been defined later in the evolution of the conceptual framework. Furthermore, each outcome was categorised only to a single outcome domain. For some outcomes, more than one domain may have been applicable, for example, 'complete tumor excision' could be categorised as a procedural event or a clinical parameter. Decisions were based on reviewers' judgement which limits the reproducibility of the exact number of outcomes categorised to each domain. Outcomes categorised as key to evaluation of innovation were, however, scrutinised in more detail by referring to the contextual information in the original publication to better understand the nature of the outcome. Furthermore, the aim of this review was to identify outcomes and domains of specific relevance to innovation and the methodology used was successful in achieving this goal. Finally, this review only included case studies of novel surgical devices. These were specifically selected to identify outcomes that may have relevance to the introduction and evaluation of medical devices, hypothesising that there may be innovation outcomes specific to device use. By focusing on devices, this review may have failed to identify key innovation outcomes of relevance to the introduction of surgical techniques not involving a device. This review, however, was only one of multiple data sources used to inform the list of outcomes for later COS development.[9 25 26]

Improvements in the processes for the introduction and evaluation of surgical innovations are a recognised need. The recent UK Independent Medicines and Medical Devices Safety Review, commissioned after the catastrophic consequences and harms experienced by some patients, highlighted the gaps and inefficiencies in the current regulatory systems.[27] Our work has explored the complexities surrounding the introduction of new procedures/devices into routine practice, highlighting shortcomings and inconsistencies in how surgical innovation is evaluated and reported.[21 22 28] This has highlighted the need for consensus on the outcomes that are essential to measure to support the safe and efficient introduction and evaluation of new surgical procedures and devices.

Findings from the current study, combined with outcomes identified from other data sources including reviews of studies of surgical procedures[25 26], interviews with surgeons and regulatory body documents, were used to inform an international Delphi consensus process with key stakeholders to agree on a core set of outcomes for use in all studies evaluating new surgical procedures/devices (the Core Outcomes for early pHasE

Surgical Innovation and deVicEs (COHESIVE) COS).[10] Mandatory measurement and reporting of this COS in all future studies of new surgical procedures and devices will ensure that key outcomes of relevance to innovation that are important to patients and key stakeholder are included. The aim of the COS is to improve standardisation and consistency across studies, allowing results to be compared and combined. Furthermore, reporting and dissemination of these outcomes in real time would inform surgeons, patients, device manufacturers and other key stakeholders of both the benefits and harms of new surgical procedures/devices in a timely manner and promote their safe, transparent and efficient evaluation and introduction and into clinical practice, avoiding patient harms. Work is currently underway to encourage the effective uptake of the new COS, to identify how to best measure these novel outcome domains (ie, metrics and timing of assessments) and to develop a real-time reporting and sharing platform to streamline the process of surgical innovation to support surgeon-innovators and improve patient outcomes.

**Author affiliations**
[1]National Institute for Health and Care Research Bristol Biomedical Research Centre, Bristol Centre for Surgical Research, Bristol Medical School: Population Health Sciences, University of Bristol, Bristol, UK
[2]Anaesthetics Department, Musgrove Park Hospital, Somerset NHS Foundation, Taunton, UK
[3]Department of Vascular Surgery, North Bristol NHS Trust, Bristol, UK
[4]Gloucestershire Hospitals NHS Foundation Trust, Gloucester, UK
[5]Division of Surgery, Bristol Royal Infirmary, University Hospitals Bristol and Weston NHS Foundation Trust, Bristol, UK
[6]Department of Gastrointestinal Surgery, North Bristol NHS Trust, Bristol, UK
[7]Bristol Dental School, University of Bristol, Bristol, UK
[8]Bristol Breast Care Centre, North Bristol NHS Trust, Westbury on Trym, UK

**Contributors** NW and RCM participated in the research design, performance of the research, data analysis and writing of the paper. CH participated in the performance of the research and data analysis. MJE, RLM and ENK participated in the performance of the research. NSB, AGKM and BGM participated in the intellectual content, research design and performance of the research. JMB participated in the intellectual content, research design and writing of the paper. KNLA and SP participated in the intellectual content, research design, performance of the research, data analysis and writing of the paper. KA and SP are guarantors for this study.

**Funding** This study was funded by the National Institute for Health and Care Research (NIHR) Biomedical Research Centre (BRC) at University Hospitals Bristol and Weston NHS Foundation Trust and the University of Bristol (BRC-1215-20011). This work was supported by the Royal College of Surgeons of England Bristol Surgical Trials Centre, the MRC ConDuCT-II (Collaboration and innovation for Difficult and Complex randomised controlled Trials In Invasive procedures) Hub for Trials Methodology Research (MR/K025643/1) and the MRC-NIHR Trials Methodology Research Partnership (TMRP). SP and AGKM are NIHR clinician scientists (NIHR CS-2016-16-019, NIHR CS-2017-17-010). NSB is an MRC clinician scientist. BGM is an NIHR clinical lecturer. JB is an NIHR senior investigator.

**Competing interests** None declared.

**Patient and public involvement** Patients and/or the public were involved in the design, or conduct, or reporting, or dissemination plans of this research. Refer to the Methods section for further details.

**Patient consent for publication** Not required.

**Provenance and peer review** Not commissioned; externally peer reviewed.

**Data availability statement** All data relevant to the study are included in the article or uploaded as supplementary information.

**ORCID iDs**
Rhiannon C Macefield http://orcid.org/0000-0002-6606-5427
Christin Hoffmann http://orcid.org/0000-0002-6293-3813
Rachael L Miller http://orcid.org/0000-0001-7918-4196
Natalie S Blencowe http://orcid.org/0000-0002-6111-2175
Angus G K McNair http://orcid.org/0000-0002-2601-9258
Barry G Main http://orcid.org/0000-0003-0622-805X
Jane M Blazeby http://orcid.org/0000-0002-3354-3330
Kerry N L Avery http://orcid.org/0000-0001-5477-2418
Shelley Potter http://orcid.org/0000-0002-6977-312X

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
