## [Reviewer comments · BMJ Open]

ARTICLE DETAILS

TITLE (PROVISIONAL)	Identification of outcomes to inform the development of a core outcome set for surgical innovation: a targeted review of case studies of novel surgical devices
AUTHORS	Wilson, Nicholas; Macefield, Rhiannon; Hoffmann, Christin; Edmondson, Matthew J; Miller, Rachael Lucia; Kirkham, Emily N; Blencowe, Natalie; McNair, Angus; Main, Barry; Blazeby, Jane; Avery, Kerry; Potter, Shelley

VERSION 1 – REVIEW

REVIEWER	Nabzyk, Christoph Mayo Clinic, Department of Anesthesia, Critical Care and Pain Medicine
REVIEW RETURNED	28-Sep-2021

GENERAL COMMENTS	The authors went through the painstaking process of characterizing the clinical evaluation processes of various medical devices. As the authors rightly state, it would be beneficial to standardize the evaluation processes for medical devices. In brief, the authors include 11 different medical technologies and provide respective device background info, a pertinent literature overview and a broad categorization of reported outcomes (figure 2). While this assessment may be important in itself, it leaves the reader with only a vague understanding of the respective processes including milestone metrics and a limited scope of possible challenges and opportunities for process improvements. For the readership it might be interesting to learn just how different the applied metrics (and possible outcomes) were for a certain technology. Maybe provide an example of great agreement between the studies pertaining to one technology versus one with very inconsistent metrics. That added granularity could further illustrate the authors' stated need for standardization of evaluation. Could the authors make metric recommendations for certain technologies they included based on their observations? Based on the general observations, do the authors suggest a certain algorithm for device evaluation? This might be a step towards a rational guide for future investigators.
---

REVIEWER	Blakely, Brette Macquarie University Faculty of Arts, Philosophy Department
REVIEW RETURNED	21-Oct-2021

GENERAL COMMENTS

Thank you for the opportunity to review this interesting and well written paper which aims to use reviewing of publications from PubMed on 11 medical devices from a range of specialties and levels of innovation to inform the development of a Core Outcome Set (COS) for surgical innovation. The paper draws attention to the issues in current reporting of surgical innovations and the need for standard innovation focused outcome measures to improve patient safety. Overall it is a very organised, clear, and concise paper. The research methods are well described. I make some suggestions below with regard to 1) strengthening the rationale for the COS by earlier mentioning the IDEAL and other reporting guidelines and the gap remaining which the COS aims to fill; 2) making more explicit that although this is contributing to a larger body of work aimed at all surgical innovation, you have deliberately limited this study to medical devices and therefore your findings reflect the breadth of surgical device innovation, not surgical innovation generally.

Abstract:
Clear and accurate.

Strengths and limitations:
See later comments in discussion

Introduction:
Pg 4 Ln 11 "One particular challenge for the evaluation of surgical innovation is the lack of standardisation in the selection, measurement and reporting of outcomes in studies of new procedures and devices, limiting data synthesis and the ability to accurately compare and contrast device efficacy [3]." The sentence is an example of the conflation around new procedures and new devices which happens at a few points in the manuscript. I will address this in more detail later when reviewing the discussion but also highlight this sentence "We hypothesised that publications reporting the introduction and evaluation of new surgical devices may generate specific insights into outcome selection and reporting in surgical innovation and potentially serve as one of several data sources to inform the development of a future COS." Pg 4 Ln 47 as needing expanding (see review of discussion for details).

Pg 4 Ln 43 "Furthermore, important 'drivers' of innovation may not be recognised and reported as outcomes in the traditional sense" Could you please provide an example or clarify what you mean here. And 'Drivers' as in goals of the innovation from a clinical perspective? Or also including other interests such as cost savings, or market domination?

I wonder why there is no mention if the IDEAL Collaboration or their recommendations with regard to best practice for the development and evaluation of surgical innovation in the introduction. This seems to me to be a major weakness in terms of contextualising the proposed COS. The rationale for the need for the innovation focused COS lacking from the IDEAL and other safety focused measures should come sooner.

Methods:
Overall the methods are very well described. Table 1 assists greatly, however some further description of the process of selecting the devices would be helpful. How many devices were

	initially identified? Could you provide examples of why some were eliminated? Was review done by all authors by consensus? For the article searches, could you clarify if the only search criteria used was the trade name? The description given suggests targeting of introduction or evaluation, however was this done at the search stage, or only at the review stage? Perhaps include with the supplementary file the search terms and dates conducted for completeness. The data extraction section is very well described as it includes a thorough discussion of how outcomes were defined and included. Results: The results are clear and concise. Key details such as quality of papers is included. Table 1 and Figure 1 are clear and display critical information. Discussion: The discussion is very important to understanding the importance and contribution of this body of work towards the larger project, as well as the gap which the COS aims to fill. As mentioned earlier in my review, I believe that the second paragraph of the discussion contains some points which should come sooner in the introduction so that the reader understands better at the beginning how this COS would fill gaps in the IDEAL and other frameworks and guidelines. The discussion also clarifies the focus on devices. I think earlier references to innovative procedures and devices should be reviewed and a more clear stance taken from the beginning. I outline below places which could be edited for clarity and suggest one further sentence in the introduction following that at Pg 4 Ln 47 to strengthen and emphasise the deliberate choice to focus on devices for this study (I imagine that choosing a technique would have been more difficult in terms of searching). You would then clarify some references to procedures and devices, for just devices; but the contribution to the overall aim would remain clear. It would also then, in my opinion, make this less of a limitation which could perhaps be addressed more lightly in the discussion as it would be incorporated into the rationale of the work at the beginning. Doing so would also prevent readers from having this concern in the back of their minds as they read the piece. Other proposed places which could then be edited for clarity on this point:  -Strengths and Limitations number 2 "Studies were chosen to reflect the breadth of surgical innovation." Should be edited to include "surgical device innovation". -Strengths and Limitations number 3 "A broad definition of an outcome was adopted to ensure that all information potentially relevant to evaluating innovation was captured." Should be edited to "evaluating device innovation". -Discussion "This review identified a broad range of outcomes identified, selected and reported in studies describing the introduction and evaluation of a purposive sample of novel
--	--

	devices, chosen to reflect the breadth of surgical innovation.” Suggest to limit to “breadth of surgical device innovation”.
REVIEWER	Selwood, Amanda Macquarie University, Australian Institute of Health Innovation
REVIEW RETURNED	31-Oct-2021
GENERAL COMMENTS	This is an important and timely paper. The analysis is well-documented and appropriate. More explanation is needed for why the authors chose a purposive sample of case studies from specialists known to the study authors, rather than using a more objective and broader search strategy for cases. However, this is a minor issue and I believe the study's conclusions still stand.

VERSION 1 – AUTHOR RESPONSE

Reviewer: 1

1. The authors went through the painstaking process of characterizing the clinical evaluation processes of various medical devices. As the authors rightly state, it would be beneficial to standardize the evaluation processes for medical devices. In brief, the authors include 11 different medical technologies and provide respective device background info, a pertinent literature overview and a broad categorization of reported outcomes (figure 2). While this assessment may be important in itself, it leaves the reader with only a vague understanding of the respective processes including milestone metrics and a limited scope of possible challenges and opportunities for process improvements.

The aim of this study was to scrutinize the published literature for a varied sample of innovative surgical device case studies to examine the types of outcomes that had been selected and measured in the reported studies, rather than characterise in full how these outcomes were used as part of the devices' clinical evaluation process. We extracted outcomes (verbatim) then conducted a conceptual exercise to classify the outcomes into outcome domains. Our intention was to identify and understand outcome domains that are specifically relevant to the evaluation of surgical device innovation, as an important first step to inform the development of a core outcome set (COS) – an agreed standardised set of outcomes to measure and report in all studies of surgical innovation. The findings from this study, combined with outcomes identified from other sources, directly informed the consensus study to develop and agree the COS which has since been published, and is referenced in the current manuscript (Avery et al. Ann Surg. 2021). In line with standard approaches for implementing core outcome sets, future work will now focus on how (i.e. using which specific metrics) to measure the outcomes in the COS. We have revised the manuscript to make this clearer and emphasise that this study identified broad outcome domains to inform the subsequent COS study and establish what outcomes to measure (Introduction; p.4, Discussion; p.10 & 11), and that future work will focus on how to measure these outcome domains, i.e. the metrics and timing of assessments (Discussion; p.13).

1. For the readership it might be interesting to learn just how different the applied metrics (and possible outcomes) were for a certain technology. Maybe provide an example of great agreement between the studies pertaining to one technology versus one with very inconsistent metrics. That added granularity could further illustrate the authors' stated need for standardization of evaluation.

We agree with the reviewer that it may be interesting to examine and compare the applied metrics between studies of the same innovative devices, and this may add to the existing evidence indicating that there is a need for standardisation in the selection, measurement and reporting of outcomes in studies of surgical innovation. The aim of our review, however, was to identify and categorise what outcomes are relevant to the evaluation of innovation to inform the development of a COS, which (as explained in 1 above) is necessary to agree on a set of outcomes that should be measured in all studies. We did not plan to compare applied metrics across the included studies in our current review. As part of future work, we will focus on operationalising the COS and consideration of appropriate metrics.

1. Could the authors make metric recommendations for certain technologies they included based on their observations? Based on the general observations, do the authors suggest a certain algorithm for device evaluation? This might be a step towards a rational guide for future investigators.

This review is part of a wider programme of work to provide recommendations on outcome selection, measurement and reporting for future investigators involved in studies of surgical innovation in the form of a COS. The intention of the current review was to undertake initial steps to explore the outcomes that have been selected and measured in existing studies of innovative surgical devices, to identify potential outcome domains particularly relevant to evaluating surgical innovation that might be relevant to include in the COS. As described in our response to point 1, the findings from this study, combined with outcomes identified from other sources, have since been used to inform the development of the COS for surgical innovation and this work is now published (Avery et al., *Ann Surg*, 2021). The COS provides recommendations and a guide for future investigators on what to measure in studies to evaluate innovative surgical procedures and devices. Planned work is now underway to determine how (i.e., using which specific metrics) to measure the COS domains. We have edited the text describing how this review was an initial step to identify a long list of potential outcomes to inform the development of the COS in the introduction (p.4&5) and the final paragraph of the discussion (p.13).

Reviewer: 2

Comments to the Author:

1. I make some suggestions below with regard to 1) strengthening the rationale for the COS by earlier mentioning the IDEAL and other reporting guidelines and the gap remaining which the COS aims to fill; 2) making more explicit that although this is contributing to a larger body of work aimed at all surgical innovation, you have deliberately limited this study to medical devices and therefore your findings reflect the breadth of surgical device innovation, not surgical innovation generally.

We thank the reviewer for these suggestions and have addressed them in detail in the comments below.

Introduction:

1. Pg 4 In 11 "One particular challenge for the evaluation of surgical innovation is the lack of standardisation in the selection, measurement and reporting of outcomes in studies of new procedures and devices, limiting data synthesis and the ability to accurately compare and contrast device efficacy [3]." The sentence is an example of the conflation around new procedures and new devices which happens at a few points in the manuscript. I will address this in more detail later when reviewing the discussion but also highlight this sentence "We hypothesised that publications reporting the introduction and evaluation of new surgical devices may generate specific insights into outcome selection and reporting in surgical

innovation and potentially serve as one of several data sources to inform the development of a future COS.” Pg 4 Ln 47 as needing expanding (see review of discussion for details). We thank the reviewer for highlighting the need to clarify this. This review deliberately focused on medical devices as one discrete data source to examine outcomes relevant to surgical device innovation. Findings were combined with a range of other data sources of relevance to surgical innovation including reviews of studies of surgical procedures, interviews with surgeons and regulatory body documents to inform the COS for surgical innovation. We have expanded the sentence in the introduction (p.5) which now reads “This study, focusing on studies of surgical devices, served as the first of multiple other data sources of relevance to surgical innovation to conceptualise outcome domains for inclusion in the COS”. We have described this further in the discussion (p.11) “This study, however, was one of many data sources to inform the list of outcomes to consider for development of the COS for surgical innovation, which used triangulated data from published literature, qualitative interviews and regulatory body documents that focused on surgical procedures and/or devices”. We have also revised the abstract conclusion (p.2) to make this clearer, as follows: “Outcome domains unique to evaluating innovative surgical devices have been identified. Findings have been combined with multiple other data sources relevant to the evaluation of surgical innovation to inform the development of a COS to measure and report in all studies evaluating novel surgical procedures/devices”.

1. Pg 4 Ln 43 “Furthermore, important ‘drivers’ of innovation may not be recognised and reported as outcomes in the traditional sense” Could you please provide an example or clarify what you mean here. And ‘Drivers’ as in goals of the innovation from a clinical perspective? Or also including other interests such as cost savings, or market domination?

We apologise that the use of this term may not be widely understood. We use the word ‘driver’ to mean a factor which causes a particular phenomenon to happen or develop. We have revised the manuscript (p.4) to clarify this “Furthermore, important ‘drivers’ (that is, factors that may influence the development) of innovation may not be recognised and reported as outcomes in the traditional sense and instead may be reported in more descriptive terms”. In the context of innovative surgical device evaluation, an example may be something that is reported descriptively in text rather than measured a priori or defined as an outcome in a study. For this reason, we adopted a broad definition of an outcome during data extraction which included any measured or reported construct or concept relating to or occurring as a result of using the device. This is described in the methods (p.6): “...a broad definition of an outcome was adopted to ensure that all information potentially relevant to the evaluation of innovation was captured. This included any measured or reported construct or concept relating to or occurring as a result of using the device. It also included descriptions or details of events and observations that occurred during the course of the study. Outcomes that were discussed (but not necessarily measured and reported as a study endpoint) by the publication’s authors as being of interest or of relevance to evaluating the device were also extracted”.

1. I wonder why there is no mention if the IDEAL Collaboration or their recommendations with regard to best practice for the development and evaluation of surgical innovation in the introduction. This seems to me to be a major weakness in terms of contextualising the proposed COS. The rationale for the need for the innovation focused COS lacking from the IDEAL and other safety focused measures should come sooner.

We thank the reviewer for this suggestion and agree that the rationale for the proposed COS would be strengthened by mentioning the IDEAL framework earlier in the manuscript. We have added the following sentence to the introduction (p.4): “The IDEAL (Idea, Development, Exploration, Assessment, Long-term Follow-up) framework describes a structure and recommended study designs for the introduction and evaluation of new surgical procedures and devices. A consensus-agreed core set of outcomes that are essential to measure at each stage of the innovation pathway, however, is lacking from the guidance”.

Methods:

1. Overall the methods are very well described. Table 1 assists greatly, however some further description of the process of selecting the devices would be helpful. How many devices were initially identified? Could you provide examples of why some were eliminated? Was review done by all authors by consensus?

We thank the reviewer for suggesting where further description would be helpful. We have revised the methods section of the manuscript (p.5) to add more information on the process of selecting the devices. We now state: "The purposive sampling strategy was discussed and agreed amongst members of the multi-disciplinary study team". Our study used a purposive sampling approach, with the aim of ensuring broad representation across several factors, including surgical specialty, disease area and the "degree of novelty" of the innovation. Unlike an objective sampling approach that is used in systematic reviews, we were not seeking to identify an exhaustive sample, but instead purposively sampled to ensure that a variety of devices were included. We have revised the manuscript in two places (p.5) to describe this in more detail as follows;

"A purposive approach was undertaken with the aim of representing, and being able to examine in depth, a broad and varied range of innovative surgical devices".

"A purposive sample of the identified devices were then selected by the study team to ensure broad representation across the surgical specialties, disease areas and "degree of novelty" of the innovation, to ensure that case studies reflected the breadth of the surgical device innovation lifecycle. Degree of novelty was categorised through discussion within the study team."

We did not formally document the total number of devices that were initially suggested by the specialists that we contacted.

1. For the article searches, could you clarify if the only search criteria used was the trade name? The description given suggests targeting of introduction or evaluation, however was this done at the search stage, or only at the review stage? Perhaps include with the supplementary file the search terms and dates conducted for completeness.

We can clarify that the only search criterion was the device trade name. Searches were performed by entering the device trade name into the PubMed database search box with no other filters or limits used. This created and ran an automatically generated search query within the PubMed database. We have now included a supplementary file that clearly describes the search process and dates (named 'Supplementary file 2_search strategy') so that this additional detail is available for readers.

Discussion:

1. The discussion is very important to understanding the importance and contribution of this body of work towards the larger project, as well as the gap which the COS aims to fill. As mentioned earlier in my review, I believe that the second paragraph of the discussion contains some points which should come sooner in the introduction so that the reader understands better at the beginning how this COS would fill gaps in the IDEAL and other frameworks and guidelines.

We thank the reviewer again for this suggestion and have included mention of the IDEAL framework and gaps in the guidelines earlier, in the introduction (p.4), as described in our response to point 7 above.

1. The discussion also clarifies the focus on devices. I think earlier references to innovative procedures and devices should be reviewed and a more clear stance taken from the beginning. I outline below places which could be edited for clarity and suggest one further sentence in the introduction following that at Pg 4 Ln 47 to strengthen and emphasise the deliberate choice to focus on devices for this study (I imagine that choosing a technique would have been more difficult in terms of searching).

You would then clarify some references to procedures and devices, for just devices; but the contribution to the overall aim would remain clear. It would also then, in my opinion, make this less of a limitation which could perhaps be addressed more lightly in the discussion as it would be incorporated into the rationale of the work at the beginning. Doing so would also prevent readers from having this concern in the back of their minds as they read the piece.

We are very grateful for this suggestion and have revised the abstract, introduction and discussion sections to make it clearer that this study focused on innovative surgical devices, and that findings were combined with studies using other data sources relevant to surgical innovation (including reviews of studies of surgical procedures, interviews with surgeons and regulatory body documents) to inform the COS development study. Specific places where we have revised the manuscript are detailed in our response to point 5 above.

1. Other proposed places which could then be edited for clarity on this point:

-Strengths and Limitations number 2 “Studies were chosen to reflect the breadth of surgical innovation.” Should be edited to include “surgical device innovation”.

-Strengths and Limitations number 3 “A broad definition of an outcome was adopted to ensure that all information potentially relevant to evaluating innovation was captured.” Should be edited to “evaluating device innovation”.

-Discussion “This review identified a broad range of outcomes identified, selected and reported in studies describing the introduction and evaluation of a purposive sample of novel devices, chosen to reflect the breadth of surgical innovation.” Suggest to limit to “breadth of surgical device innovation”.

We thank the reviewer for these suggestions and have edited the manuscript accordingly to specify “surgical device innovation” where suggested and in other places throughout the manuscript.

Reviewer: 3

1. More explanation is needed for why the authors chose a purposive sample of case studies from specialists known to the study authors, rather than using a more objective and broader search strategy for cases. However, this is a minor issue and I believe the study's conclusions still stand.

We thank the reviewer for this comment, which was also raised by Reviewer 2. Please see our response to point 8 above which describes where we have revised the manuscript to explain the methods for sampling case studies in more detail.

VERSION 2 – REVIEW

REVIEWER	Nabzyk, Christoph Mayo Clinic, Department of Anesthesia, Critical Care and Pain Medicine
REVIEW RETURNED	31-Jan-2022

GENERAL COMMENTS	the authors provide a well articulated and balanced response to reviewers and provide focuses edits to the manuscript that further improve its quality.
---

REVIEWER	Blakely, Brette Macquarie University Faculty of Arts, Philosophy Department
REVIEW RETURNED	07-Feb-2022

GENERAL COMMENTS

Thank you for the opportunity to review a revised version of this manuscript. The reader's ability to appreciate the context and purpose of the work has been improved by the revisions undertaken by the authors. This is an interesting paper and contributes an important piece to the growing body of work looking to improve the safety of surgical innovation.